# Improved Retinex-Theory-Based Low-Light Image Enhancement Algorithm

**Jiarui Wang [1], Hanjia Wang [1], Yu Sun [1] and Jie Yang [2,***

1   Dalian Naval Academy, Dalian 116013, China
2   College of Information Science and Engineering, Northeastern University, Shenyang 110819, China
*   Correspondence: yangjie@ise.neu.edu.cn

**Abstract:** Researchers working on image processing have had a hard time handling low-light images due to their low contrast, noise, and brightness. This paper presents an improved method that uses the Retinex theory to enhance low-light images, with a network model mainly composed of a Decom-Net and an Enhance-Net. Residual connectivity is fully utilized in both the Decom-Net and Enhance-Net to reduce the possible loss of image details. Additionally, Enhance-Net introduces a positional pixel attention mechanism that directly incorporates the global information of the image. Specifically, Decom-Net serves to decompose the low-light image into illumination and reflection maps, and Enhance-Net serves to increase the brightness of the illumination map. Finally, via adaptive image fusion, the reflectance map and the enhanced illuminance map are fused to obtain the final enhanced image. Experiments show better results in terms of both subjective visual aspects and objective evaluation indicators. Compared to RetinexNet, the proposed method shows improvements in the full-reference evaluation metrics, including a 4.6% improvement in PSNR, a 1.8% improvement in SSIM, and a 10.8% improvement in LPIPS. Additionally, it achieved an average improvement of 17.3% in the no-reference evaluation metric NIQE.

**Keywords:** image processing; low-light image enhancement; Retinex theory; low/normal-light image





## 1. Introduction

Low-light image enhancement is an important branch of image processing, which is mainly applied in fields such as photo surveillance, intelligent driving, and military applications [1]. The main causes of low-illumination images are low light at night and backlighting. Images produced in such environments are generally characterized by low contrast and high noise, which seriously affect the amount of information that can be reflected in the image. In certain circumstances, low-light images can even become distorted, and black can completely obscure the original image information [2–4]. Therefore, restoring the brightness of low-light images and improving image contrast is a meaningful research topic.

Various algorithms have been created for processing low-light images. There are three main types of traditional methods for enhancing images at low light: image defogging, the histogram-based method, and the Retinex-based method. These methods are not very effective in processing images, highlighting problems such as color distortion and noise amplification. Therefore, it is of utmost importance to constrain image noise and equalize image color while enhancing image brightness and contrast, which is a hot topic of research in the field of low-light image enhancement (Figure 1).

The field of image processing has been greatly impacted by deep learning. Research is needed to develop it. It has been observed that certain methods that combine the techniques of deep learning and the Retinex framework for low-light image enhancement have been proven to be effective, such as RetinexNet [5] and R2RNet [6]. However, existing methods suffer from problems such as amplified noise, color distortion, and loss of image details

after enhancement. Therefore, we propose a new method for extracting images from the network using an attention mechanism that is different from deep learning and the Retinex theory. Furthermore, we introduce a color consistency loss function to the other loss functions to further improve the image quality. The proposed method's flowchart can be seen in Figure 2. The main contributions of this paper are as follows:

(1) We designed a new decomposition module with the introduction of residual connections, aiming to reduce the loss of detailed information during the decomposition process and obtain more accurate illumination and reflection maps.

(2) An effective method is proposed for low-illumination image enhancement.

(3) By introducing an attention mechanism to improve the network structure and adding a color consistency loss function, the illumination and reflection images can be effectively obtained via the processing.

(4) The proposed method is significantly better than the current techniques for improving low-light images when it comes to visual perception and evaluation indexes.

The paper begins with Section 2, which covers the various works related to the Retinex theory. In Section 3, we introduce the proposed model, as well as the loss functions that are utilized. In Section 4, we conduct experimental evaluations, including subjective assessments. The paper concludes in Section 5.

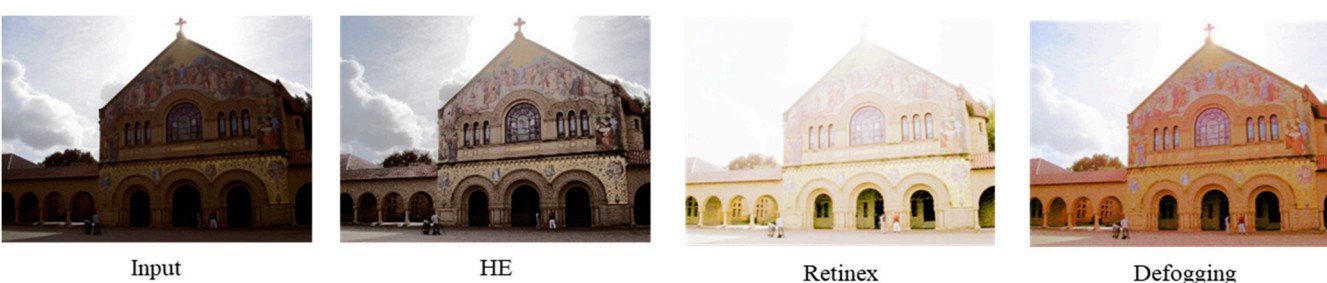

**Figure 1.** Enhanced effect of the traditional methods.

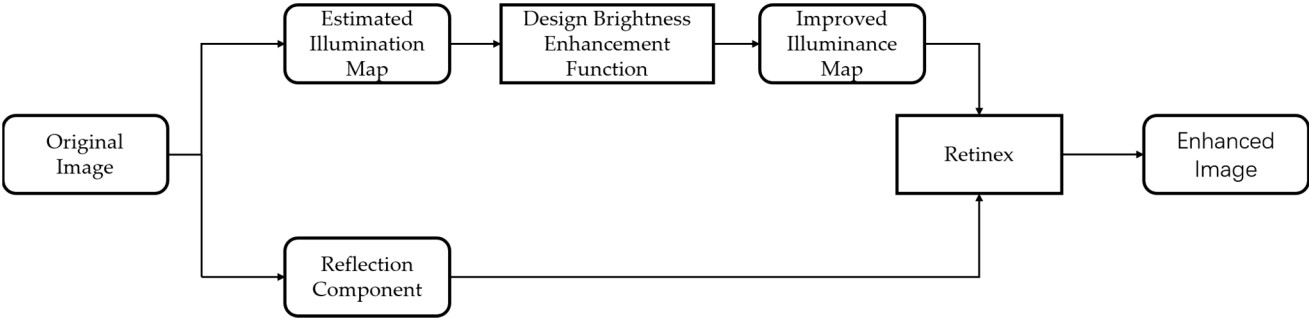

**Figure 2.** A flowchart of the proposed method.

## 2. Related Work

Numerous techniques have been proposed for enhancing low-light images. Algorithms based on histogram equalization have derived dynamic histogram equalization (HE), adaptive histogram equalization (AHE) [7], and contrast-limited adaptive histogram equalization (CLAHE) [8], among others. HE is a process that increases the contrast of an image by distributing its gray range more evenly. AHE divides the image into several small blocks for histogram equalization processing, which can better improve the local contrast of the image and achieve better enhancement results. CLAHE sets a threshold and only averages the parts exceeding the threshold to each gray level, eliminating the blocky effect that may be caused by AHE. Traditional Retinex-based methods mainly include the single-scale Retinex method (SSR), multi-scale Retinex method (MSR) [9], and MSRCP [10], among

others. SSR is carried out by using filters, which are designed to filter the various channels of an image. They are then used to create the illumination image. The low-light image and the illumination image are logarithmically processed and subtracted to achieve the enhancement effect. MSR is equivalent to stacking multiple SSRs, which can be regarded as the weighted sum of multiple SSRs. MSRCP adds color balance, normalization, etc., to the multi-scale MSR results. Low-light image enhancement based on dehazing applies the dehazing algorithm to the reflection image of the low-light image, and after processing the image, it is inverted to obtain the final enhancement effect.

LLNet [11] presents a method that trains a self-encoder using a synthetic image dataset. It achieves noise reduction while still enhancing the image. Fu et al. [12] proposed a network that aims to enhance low-light images using an unsupervised model known as LE-GAN. The network uses a light-aware attention module to reduce the noise and extract various features. They also introduced a loss function that can be used to address the issue of overexposure. Ma et al. [13] proposed a self-calibrating lighting framework that features a lighting learning process with crowd-share for enhancement purposes. In contrast, the proposed self-calibrating lighting framework can significantly reduce computational costs and achieve fast processing. Wei et al. proposed RetinexNet, which combines the Retinex theory with convolutional neural networks to enhance low-light images by enhancing the illumination image and reducing noise in the reflectance image using BM3D. R2RNet [6] is also based on Retinex and deep learning networks, and it designs a residual module (RM) and adds it to its Decom-Net, noise reduction network, and augmentation network to obtain better component results. Some algorithms combine Retinex with zero-shot networks, such as RRDNet [14]. RRDNet is modeled on the theory of Retinex and features the method by decomposing the low-illumination image into illumination, reflection, and noise images, thus achieving noise reduction. The method converts the Decom-Net into a generative network, thus reducing the coupling between the illuminance and reflection images.

The key to the Retinex method is to obtain higher quality illumination and reflection images. The residual network has shown remarkable performance in various image processing tasks due to its unique jump connection structure, which effectively prevents the gradient from exploding or disappearing during the training of convolutional neural networks. We build on this foundation by integrating the residual network into the decomposition network to obtain better decomposition results. Furthermore, we introduce an improved color invariant loss function for enhancing the output. Figure 3 demonstrates the decomposition and enhancement results of the shimmering images.

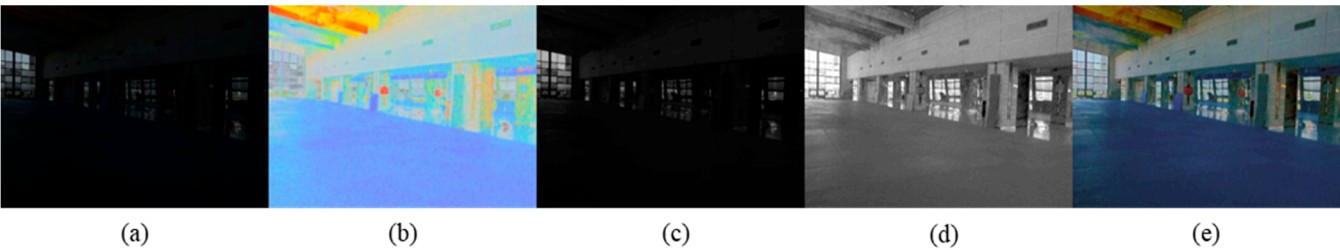

**Figure 3.** The decomposition results obtained by our method: (**a**) the low-light image, (**b**) the reflectance image, (**c**) the illumination image, (**d**) the enhanced illumination image, and (**e**) the enhanced result.

## 3. Proposed Method

This paper proposes a method that fully considers the image decomposition and the image quality degradation brought about by the enhancement process. The model diagram of this method is shown in Figure 4, which is divided into a Decom-Net and an Enhance-Net. The low-illumination image is used as the input of the Decom-Net, and the output of the Decom-Net is the illumination image and the reflection image. The decomposed illuminance and reflection images are fed into the Enhance-Net to obtain the enhanced

illuminance image. Finally, the decomposed reflection image and the enhanced illuminance image are fused to obtain the final enhancement result.

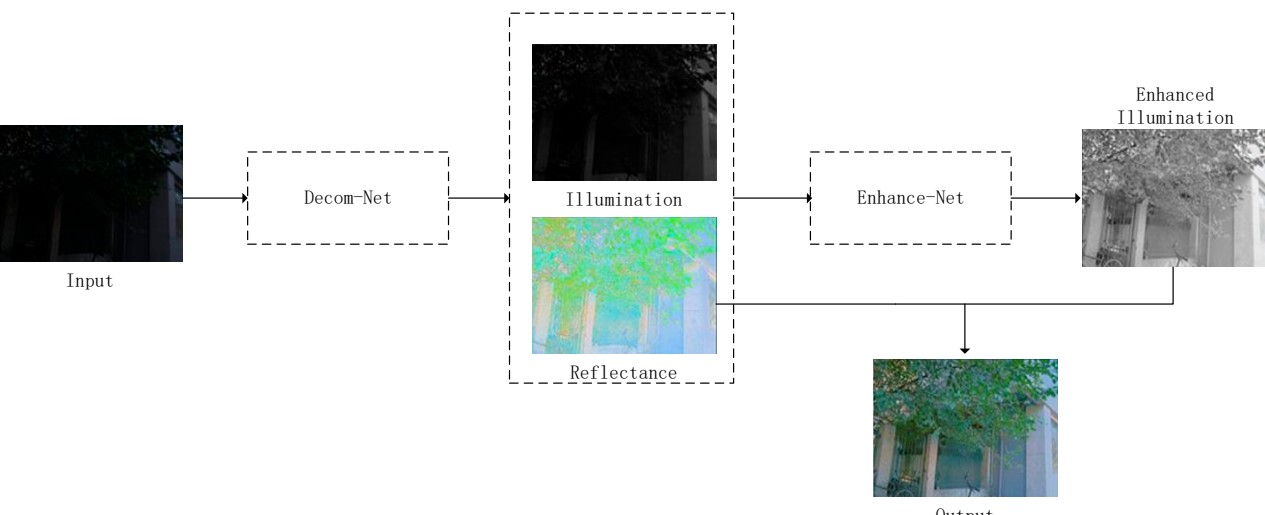

**Figure 4.** Flow chart of the algorithm in this paper.

### 3.1. Decom-Net

The key to the Retinex theory lies in obtaining high-quality illumination and reflection images, which in turn greatly affect the final enhancement results. Therefore, designing a high-quality decomposition network is one of the main focuses of this paper. The design of residual networks aims to solve the problem of gradient explosion or vanishing that may occur when the depth of convolutional models increases. Stacked layers, which include pooling and convolutional techniques, can result in a loss of detail during the processing of images. The proposed network avoids this issue by having a skip-connection structure. This can help minimize the effects of neural networks on the image quality. The decomposition network proposed in this paper is illustrated in Figure 5, where S denotes the input image, R denotes the reflection image, and I denotes the illumination image. A $1 \times 1$ convolutional layer is added as a connecting layer in the decomposition network. During the training process, with the guidance of sharing the reflection component, the illumination and reflection images learn to be decomposed from the paired image dataset containing low-light images and truth images.

### 3.2. Enhance-Net

The Enhance-Net architecture proposed in this paper is illustrated in Figure 6. In order to adjust the local brightness of the illumination map while maintaining consistency in the global illumination, we have utilized multi-scale connection methods extensively. Furthermore, we have introduced the positional pixel attention mechanism, which weighs the positional information of the pixels in the image at a pixel level. Unlike the pixel-level attention mechanism, the positional pixel attention mechanism places greater emphasis on the attention weight of each position. This enhances the robustness and generalization ability of the model while also smoothing out some noise and having a certain denoising effect. It is worth noting that the network also achieved high results in image separation, target classification, and other directions.

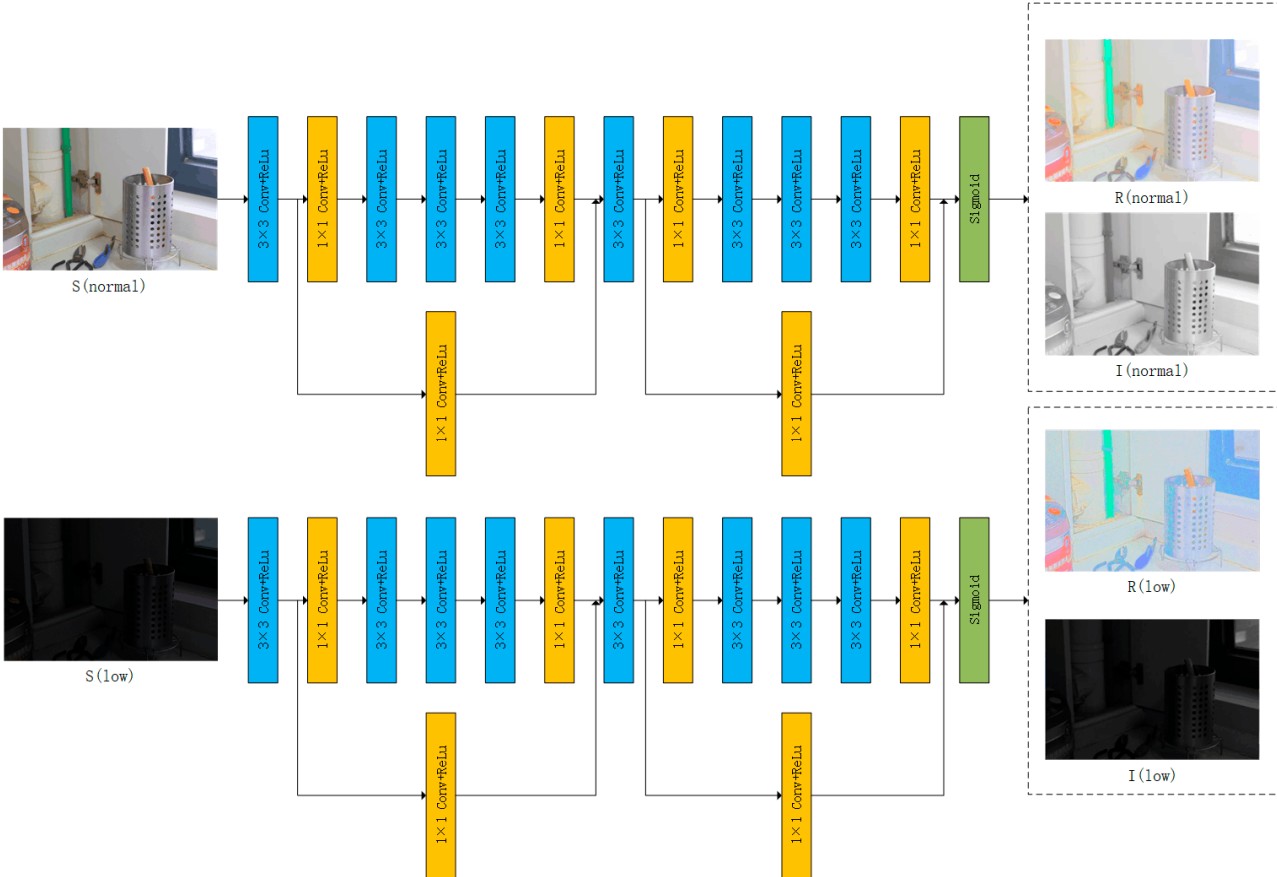

**Figure 5.** Structure of the Decom-Net.

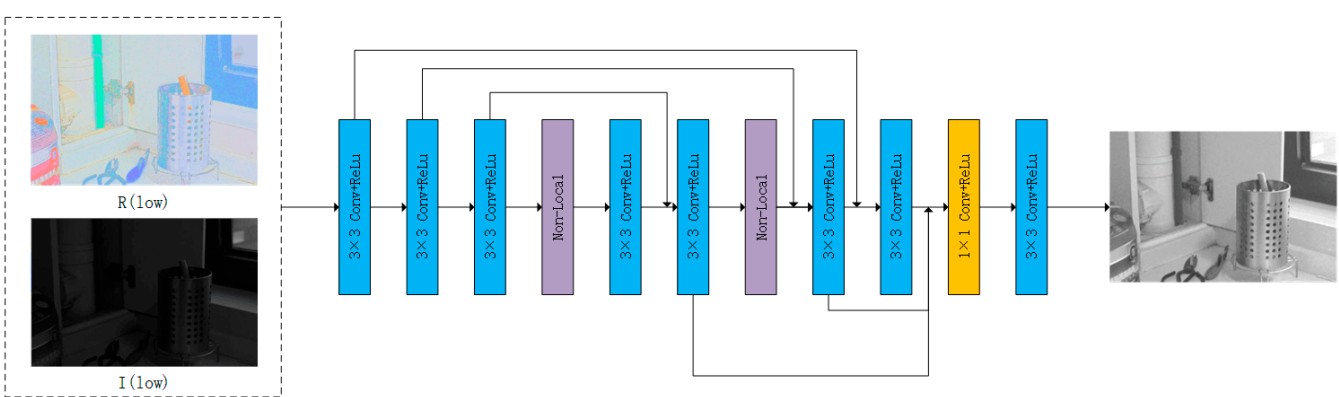

**Figure 6.** Structure of the Enhance-Net.

### 3.3. Loss Function

To further improve the decomposition and enhancement effects, this paper adopts a series of loss functions. We will describe each of these loss functions in detail.

The decomposition loss function $L_{Decom}$ consists of three components: the reconstruction loss, the reflectance component consistency loss, and the illumination smoothness loss. The formula for the decomposition loss function is as follows:

$$L_{Decom} = L_{re} + \lambda_1 L_r + \lambda_2 L_i \tag{1}$$

where $L_{re}$ denotes reconstruction loss, $L_r$ denotes reflection component consistency loss, and $L_i$ denotes illumination smoothness loss. $\lambda_1$ and $\lambda_2$ denote the coefficients of reflection component loss and illumination component loss, respectively.

Based on the decomposition of the low-light image into illumination and reflectance maps, as well as the illumination and reflectance maps of a normal-light image, this paper proposes the following formula for image reconstruction:

$$L_{re} = \lambda_3 \sum_{i=1}^{N} \parallel R_n \circ I_l - S_l \parallel_1 + \lambda_4 \sum_{i=1}^{N} \parallel R_l \circ I_n - S_n \parallel_1 + \sum_{i=1}^{N} \parallel R_l \circ I_l - S_l \parallel_1 + \sum_{i=1}^{N} \parallel R_n \circ I_n - S_n \parallel_1 \tag{2}$$

where $\lambda_3$ and $\lambda_4$ represent their respective coefficients. $S_n$, $S_l$, $R_n$, $I_n$, $R_l$, and $I_l$ represent the normal-light image, low-light image, reflection and illumination maps of the normal-light image, and reflection and illumination maps of the low-light image, respectively.

The reflection component consistency loss is defined as follows:

$$L_r = \parallel R_l - R_n \parallel_1 \tag{3}$$

The illumination smoothness loss is a minimization model that weights the gradient of the reflectance map, and its expression is as follows:

$$L_i = \sum_i \parallel \nabla I_i \circ \exp\left(-\lambda_g \nabla R_i\right) \parallel_1 \tag{4}$$

where $\nabla$ denotes the horizontal and vertical gradients of the image, and $\lambda_g$ is the balance coefficient.

The enhancement loss function consists of the relighting loss and the color constancy loss, which are expressed as follows:

$$L_{En} = L_{light} + \lambda_5 L_{col} \tag{5}$$

where $L_{En}$ denotes enhancement loss, $L_{col}$ denotes color constancy loss, and $\lambda_5$ is the coefficient of noise reduction loss and color constancy loss.

$$L_{light} = \parallel R_l * \hat{I} - S_n \parallel_1 \tag{6}$$

where $\hat{I}$ is the output of the Enhance-Net.

The color constancy loss is based on the assumption of the grey world color constancy theory, which assumes that the average color of each sensor channel in an image is gray. The goal of the color constancy loss process is to eliminate any biases that may appear in the enhanced image. The equation for the color constancy loss is as follows:

$$L_{col} = \sum_{\forall (p,q) \in Y} (T^p - T^q)^2, Y \in \{(R,G), (R,B), (G,B)\} \tag{7}$$

where $T^p$ denotes the average intensity value of the $p$ channels in the enhanced image method.

## 4. Experiments

### 4.1. Experimental Implementation Details

The method adopted is trained and tested on an NVIDIA GeForce RTX2080 graphics card, combined with an Intel i7-8700 processor and 32G of RAM. The software environment employed Pycharm version 2022, equipped with the Anaconda Python 3.7 interpreter and the Pytorch 1.12.1 framework. The initial learning rate was set to 0.001, with $\lambda_1$ and $\lambda_2$ both set to 0.1 in Equation (1), $\lambda_3$ and $\lambda_4$ set to 0.01 in Equation (2), $\lambda_5$ set to 0.01 in Equation (5), and the remaining weights set to 1.

The training dataset utilized in this study comprises the LOL dataset [5] and a partially synthesized dataset. The total number of images included in the LOL dataset was 500 pairs. Out of these pairs, 485 were used as training sets, and 15 were used as test sets. This is re-

garded as the most commonly used image enhancement dataset in the research community. For the synthetic version of the dataset, we collected about 1000 images to compensate for the low number of LOL files. The LOL dataset is obtained by capturing images using a camera with varying exposure times and ISO sensitivity. The dataset primarily consists of indoor images with a resolution of $400 \times 600$. The synthetic dataset is generated by processing the RAISE image dataset using Adobe Lightroom. The images in this dataset are unprocessed high-resolution RAW images captured by four photographers over a period of three years using three cameras at different locations across Europe, capturing various scenes at different times.

We compare quantitatively with traditional methods Retinex, MSRCP, and deep-learning-based methods such as MBLLEN [15], lightenNet [16], RetinexNet [5], Enlighten-GAN [17], RRDNet [14], DSLR [18], ExCNet [19], Zero-DCE, Zero-DCE++ [20], and R2RNet. For the testing dataset, we used the paired LOL dataset, VE-LOL-H dataset, and unpaired LIME dataset. We used reference-based evaluation metrics including peak signal-to-noise ratio (PSNR) [21], structural similarity (SSIM) [21], mean squared error (MSE), learned perceptual image patch similarity (LPIPS), and variance inflation factor (VIF) [22], as well as reference-free natural image quality evaluation metrics such as the Natural Image Quality Evaluator [23] to evaluate the deep-learning-based methods.

PSNR is a supervised evaluation method that measures the quality of an image's reconstruction. According to the evaluation, the closer two images are to one another, the better the reconstruction's quality.

The structural similarity of an image is measured by using SSIM, which takes into account the differences in structure, contrast, and brightness between different images. Values range from zero to 1, which indicates a better-quality image.

MSE is an image evaluation metric that calculates the average of the squared differences between image pixels and reference image pixels. A smaller MSE value indicates that the image quality is closer to that of the reference image.

LPIPS is a statistical measure of the perceptual similarity of two images. It is commonly used to evaluate the human eye's perception of images. Compared to other indices, the LPIPS is closely related to the subjective evaluation of humans.

VIF is an index that combines the statistical and natural data collected by the visual system and the human eye to evaluate the quality of images. It shows a correlation between the two values and the images' perceived visual quality.

NIQE is a non-reference evaluation metric. Both the PSNR and SSIM datasets are paired, but we need non-reference evaluation metrics for non-paired image datasets. Furthermore, it has been found that the PSNR and SSIM metrics do not apply to image texture detail, and NIQE can effectively bridge this gap. The smaller the value of NIQE, the better the image quality.

### 4.2. Experimental Comparison

We compare the methods in this paper visually and objectively with traditional enhancement methods and the latest enhancement methods. Figures 7–9 show the comparison results of the aforementioned methods on the LOL dataset, VE-LOL-H dataset [24], and LIME dataset [25].

Figure 7 illustrates the effect of different methods on the enhancement of images from the LOL dataset. The original input image in low illumination has very low brightness and significant loss of image information. From Figure 7, it is apparent that some of the methods, such as LightenNet, RRDNet, and DSLR, do not show a significant enhancement effect since the enhanced image is not fully exposed, and the enhancement effect could be better. The color of the images from Retinex, RetinexNet, MBLLEN, and EnlightenGAN is distorted, especially in the case of EnlightenGAN in which the color of the leaves turns yellow, and the color deviation is significant. In contrast, the proposed method was able to enhance the image quality significantly. It also preserved the details of the image.

Figure 8 illustrates the enhancement of indoor images from the VE-LOL-H dataset by different methods. The brightness enhancement of the two conventional methods, Retinex and MSRCP, is more pronounced in Figure 8; however, image noise is amplified, and severe color distortion is present. The brightness enhancement effect of Zero-DCE and Zero-DCE++ is better than the method proposed in this paper, but the overall image is obviously white, and the image brightness is too saturated.

Figure 9 illustrates the results of enhancing images from the LIME dataset, wherein an image with a backlighting effect was selected. The original image exhibits rich color information, while some methods introduced severe distortion in the backlight region. For instance, the Retinex effect resulted in the complete disappearance of white clouds. Furthermore, the white clouds in the background of the DSLR enhancement were considerably attenuated, RetinexNet and MSRCP resulted in a noticeable amplification of noise, and the enhanced images of Zero-DCE and Zero-DCE++ exhibited excessive whiteness and severe color deviation. In contrast to the aforementioned methods, the proposed approach achieved a more balanced enhancement between color brightness and contrast.

This study compares four image datasets, including the LOL [5], LIME [25], VV, and DICM [26] datasets. To ensure a fair comparison, we selected publicly available codes without any modifications, and training datasets were also sourced from the LOL dataset. For methods that use unpaired images for training, such as EnlightenGAN and Zero-DCE, we compared them using pre-trained models provided by their respective authors. We evaluated the LOL dataset's test set using objective evaluation metrics, such as PSNR, SSIM, MSE, and LPIPS. The specific results are shown in Table 1. The LIME, VV, and DICM datasets only contain low-light images. We evaluated the image quality of these datasets using NIQE, and the specific results are shown in Table 2.

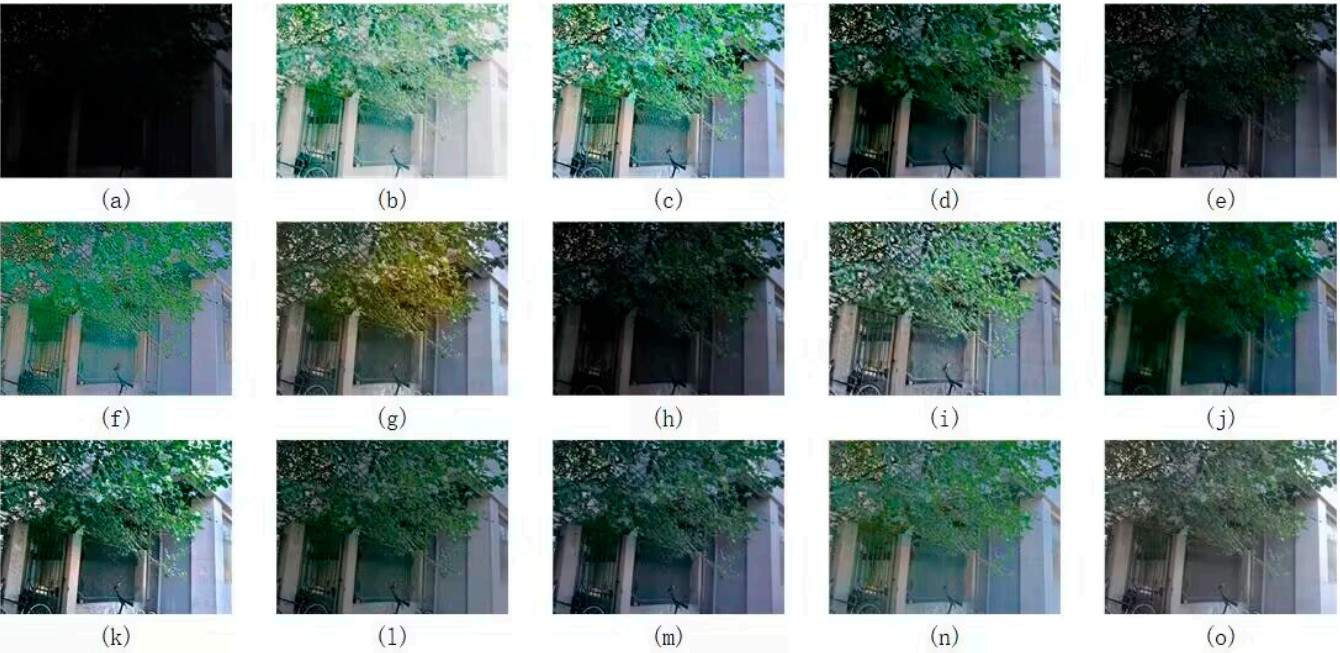

**Figure 7.** The results of a low-light image enhancement procedure on the LOL dataset were obtained. (**a**) Low-illumination image; (**b**) Retinex; (**c**) MSRCP; (**d**) MBLLEN; (**e**) LightenNet; (**f**) RetinexNet; (**g**) EnlightenGAN; (**h**) RRDNet; (**i**) R2RNet; (**j**) DSLR; (**k**) ExCNet; (**l**) Zero-DCE; (**m**) Zero-ECD++; (**n**) our method; (**o**) Truth value.

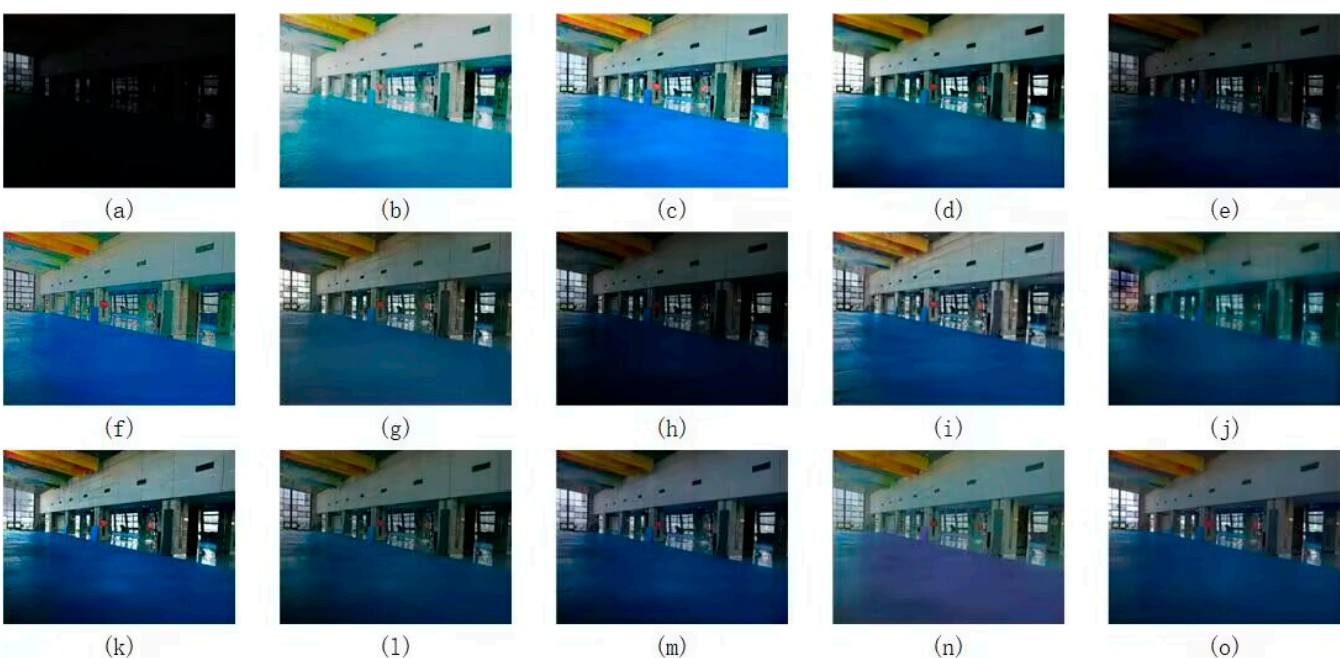

**Figure 8.** The results of a low-light image enhancement procedure on the VE-LOL-H dataset were obtained. (**a**) Low-illumination image; (**b**) Retinex; (**c**) MSRCP; (**d**) MBLLEN; (**e**) LightenNet; (**f**) RetinexNet; (**g**) EnlightenGAN; (**h**) RRDNet; (**i**) R2RNet; (**j**) DSLR; (**k**) ExCNet; (**l**) Zero-DCE; (**m**) Zero-ECD++; (**n**) our method; (**o**) Truth value.

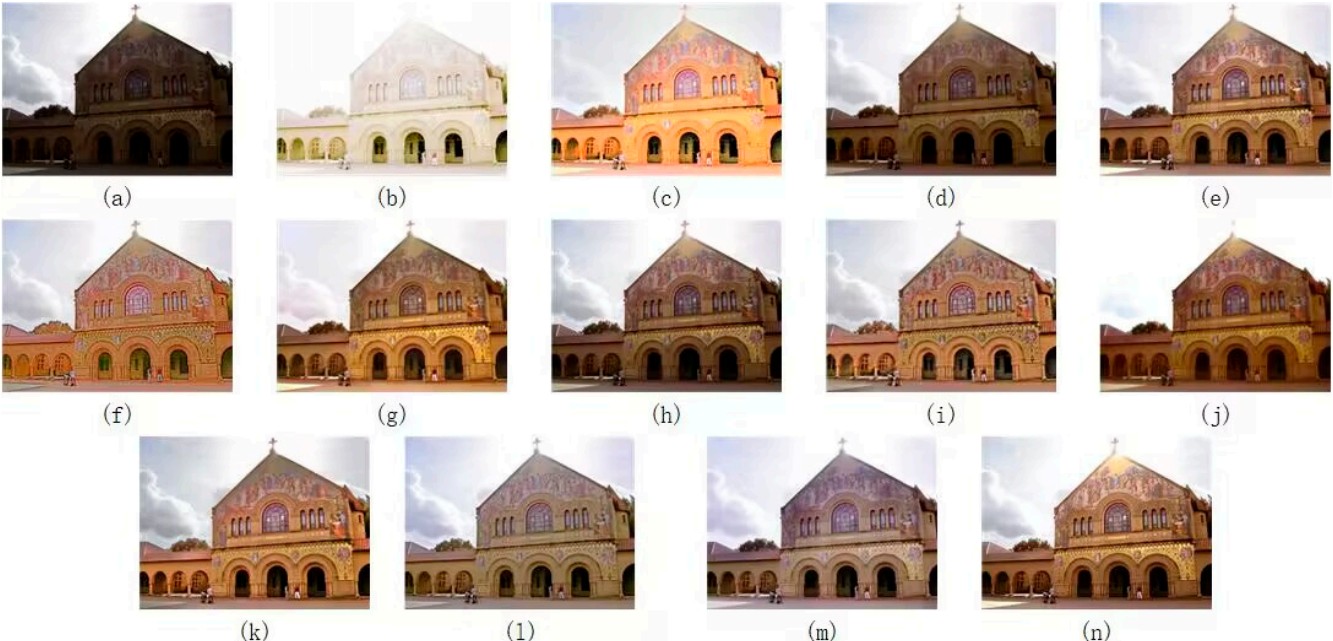

**Figure 9.** The results of a low-light image enhancement procedure on the LIME dataset were obtained. (**a**) Low-illumination image; (**b**) Retinex; (**c**) MSRCP; (**d**) MBLLEN; (**e**) LightenNet; (**f**) RetinexNet; (**g**) EnlightenGAN; (**h**) RRDNet; (**i**) R2RNet; (**j**) DSLR; (**k**) ExCNet; (**l**) Zero-DCE; (**m**) Zero-ECD++; (**n**) our method.

**Table 1.** Quantitative comparisons were conducted on the LOL dataset using PSNR, SSIM, MSE, and LPIPS as evaluation metrics. The best results are highlighted in bold and underlined. The second-best results are in italics, and the third-best results are in bold.

| Method | PSNR | SSIM | MSE | LPIPS | VIF |
|---|---|---|---|---|---|
| Input | 7.773 | 0.121 | 12613.620 | 0.560 | 0.266 |
| LightenNet [16] | 11.965 | 0.632 | 5661.574 | 0.359 | 0.704 |
| MBLLEN [15] | **17.655** | *0.930* | **1515.911** | *0.287* | **0.956** |
| RetinexNet [5] | 17.399 | 0.916 | 1565.746 | 0.363 | <u>**0.963**</u> |
| EnlightenGAN [17] | 17.539 | **0.919** | 1960.018 | 0.326 | 0.938 |
| DSLR [18] | 14.935 | 0.838 | 3679.389 | 0.403 | 0.862 |
| ExCNet [19] | 16.390 | 0.908 | 2159.039 | 0.327 | 0.935 |
| RRDNet [14] | 10.878 | 0.511 | 6966.992 | 0.336 | 0.642 |
| Zero-DCE [27] | 14.861 | 0.827 | 3281.716 | 0.335 | 0.868 |
| Zero-DCE++ [20] | 14.682 | 0.855 | 3207.455 | 0.340 | 0.882 |
| R2RNet [6] | *18.179* | 0.892 | *1498.952* | <u>0.272</u> | 0.950 |
| Our method | <u>**18.208**</u> | <u>**0.932**</u> | <u>**1459.655**</u> | 0.318 | *0.959* |

**Table 2.** Comparison of the LIME, VV, and DICM datasets was conducted in terms of NIQE. The best results are highlighted in bold and underlined. The second-best results are in italics, and the third-best results are in bold.

| Method | LIME | VV | DICM | Avg. |
|---|---|---|---|---|
| LightenNet [16] | 4.681 | 3.729 | 3.735 | 4.048 |
| MBLLEN [15] | 4.818 | 4.294 | 3.442 | 4.185 |
| RetinexNet [5] | 4.361 | 3.422 | 4.209 | 3.997 |
| EnlightenGAN [17] | *3.697* | <u>2.981</u> | 3.570 | **3.416** |
| DSLR [18] | 4.033 | 3.649 | 3.389 | 3.690 |
| ExCNet [19] | 4.162 | 3.783 | *3.039* | 3.661 |
| RRDNet [14] | 4.522 | 3.845 | 3.992 | 4.120 |
| Zero-DCE [27] | 3.928 | **3.217** | 3.716 | 3.620 |
| Zero-DCE++ [20] | 3.843 | 3.341 | <u>2.835</u> | <u>3.339</u> |
| R2RNet [6] | **3.704** | *3.093* | 3.503 | 3.433 |
| Our method | <u>3.547</u> | 3.364 | **3.312** | *3.408* |

Table 1 shows the comparison results of the PSNR, SSIM, MSE, and LPIPS metrics for the LOL dataset's test set. From this, we can see that all brightness enhancement methods improved the performance metrics. The PSNR, SSIM, and MSE metrics of this method all scored high, with the PSNR metric leading the second place by 0.029 dB, the SSIM metric leading the second place by 0.002, and the MSE metric leading the second place by 39.287. The LPIPS and VIF metrics were also ranked in third and second place, respectively. This indicates that the enhancement effect of this method is closer to the real image than other methods.

To evaluate the LIME, VV, and DICM datasets without reference evaluation indicators, we used the NIQE indicator as shown in Table 2. Our method ranks in the top three positions for both the LIME and DICM datasets. For a fairer comparison, we calculated the average scores of the three datasets and found that our proposed method outperforms most existing methods.

The results of the paper's experimental procedures exhibited good scores in both the objective indicator and subjective visual evaluations. The proposed method's superiority was also highlighted. However, we also need to point out that there is still a slight color distortion phenomenon in comparison with some control images, although it is not very obvious, and this is still a direction for the further development of this work.

In Table 3, we calculated the computational complexity based on the average runtime in the LOL test set. It can be observed that the runtime varies significantly among different methods, with EnlightenGAN being the fastest and RRDNet being slower. Compared to

other methods, the proposed approach also outperforms the majority of them. However, achieving real-time enhancement for videos remains a key focus for future research in this field.

**Table 3.** The average running time of the proposed method for enhancing images in the LOL dataset. The best results are highlighted in bold and underlined. The second-best results are in italics, and the third-best results are in bold.

| Method | Running Time (s) |
|---|---|
| LightenNet [16] | 0.983 |
| MBLLEN [15] | 3.975 |
| RetinexNet [5] | *0.469* |
| EnlightenGAN [17] | **0.012** |
| DSLR [18] | 0.942 |
| ExCNet [19] | 12.748 |
| RRDNet [14] | 78.386 |
| Zero-DCE [27] | 0.671 |
| Zero-DCE++ [20] | 0.747 |
| R2RNet [6] | **3.704** |
| Our method | **0.633** |

### 4.3. Ablation Study

In order to demonstrate the accuracy of the loss function in our parameter settings, we conducted ablation experiments by experimenting with the values of $\lambda_1$, $\lambda_2$, $\lambda_3$, and $\lambda_4$ to investigate their impact on enhancing the images, as shown in Figure 10. From the results, we found that there were significant differences between the compared images, and some of the images had significant color deviation, such as the green book part of some images showing a yellowish hue. To better select experimental parameters, we evaluated the comparison results of various parameters using metrics, as shown in Table 3.

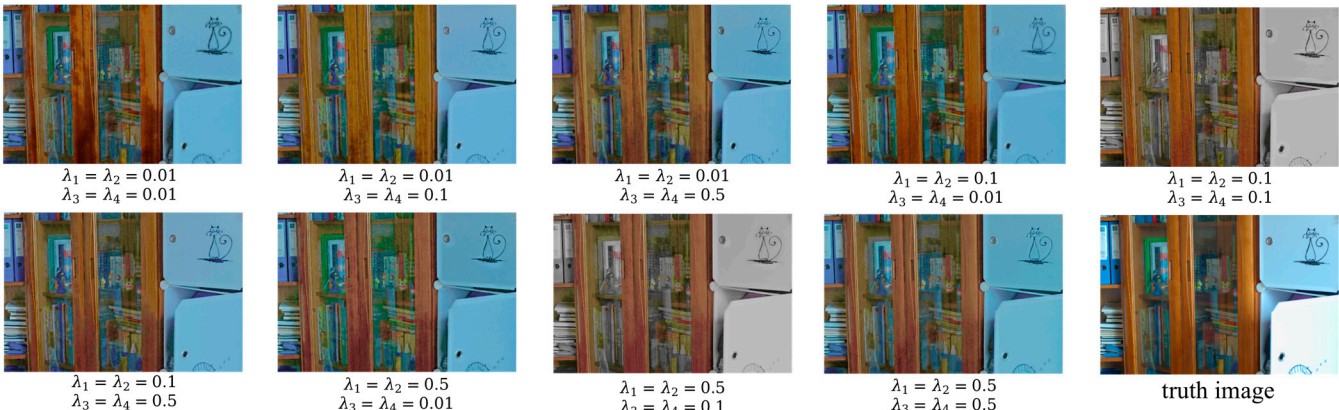

$\lambda_1 = \lambda_2 = 0.01$
$\lambda_3 = \lambda_4 = 0.01$    $\lambda_1 = \lambda_2 = 0.01$
$\lambda_3 = \lambda_4 = 0.1$    $\lambda_1 = \lambda_2 = 0.01$
$\lambda_3 = \lambda_4 = 0.5$    $\lambda_1 = \lambda_2 = 0.1$
$\lambda_3 = \lambda_4 = 0.01$    $\lambda_1 = \lambda_2 = 0.1$
$\lambda_3 = \lambda_4 = 0.1$

$\lambda_1 = \lambda_2 = 0.1$
$\lambda_3 = \lambda_4 = 0.5$    $\lambda_1 = \lambda_2 = 0.5$
$\lambda_3 = \lambda_4 = 0.01$    $\lambda_1 = \lambda_2 = 0.5$
$\lambda_3 = \lambda_4 = 0.1$    $\lambda_1 = \lambda_2 = 0.5$
$\lambda_3 = \lambda_4 = 0.5$    truth image

**Figure 10.** Effect of different parameter values on the enhanced images.

According to the subjective visual comparison in Figure 10, it can be observed that the enhancement effect is closest to the reference image when $\lambda_1 = \lambda_2 = 0.1$ and $\lambda_3 = \lambda_4 = 0.01$. According to Table 3, the best results in terms of the SSIM, MSE, and NIQE metrics are achieved when $\lambda_1 = \lambda_2 = 0.1$ and $\lambda_3 = \lambda_4 = 0.01$. The algorithm also performs well in terms of PSNR, ranking among the top three. Based on the subjective and objective evaluations from Figure 10 and Table 4, we choose the parameter settings of $\lambda_1 = \lambda_2 = 0.1$, $\lambda_3 = \lambda_4 = 0.01$.

**Table 4.** The results of the evaluation were compared with the different parameters using the NIQE, MSE, PNSR, and SSIM metrics. The best results are highlighted in bold and underlined. The second-best results are in italics, and the third-best results are in bold.

| Method | PSNR | SSIM | MSE | NIQE |
|---|---|---|---|---|
| $\lambda_1 = \lambda_2 = 0.01, \lambda_3 = \lambda_4 = 0.01$ | 17.466 | *0.931* | 1660.814 | **3.404** |
| $\lambda_1 = \lambda_2 = 0.01, \lambda_3 = \lambda_4 = 0.1$ | <u>**17.590**</u> | 0.928 | **1624.519** | 3.560 |
| $\lambda_1 = \lambda_2 = 0.01, \lambda_3 = \lambda_4 = 0.5$ | 17.281 | 0.927 | 1703.411 | *3.282* |
| $\lambda_1 = \lambda_2 = 0.1, \lambda_3 = \lambda_4 = 0.01$ | **17.470** | <u>**0.932**</u> | <u>**1592.601**</u> | <u>**3.184**</u> |
| $\lambda_1 = \lambda_2 = 0.1, \lambda_3 = \lambda_3 = 0.1$ | 17.087 | 0.913 | 1850.028 | 3.931 |
| $\lambda_1 = \lambda_2 = 0.1, \lambda_3 = \lambda_4 = 0.5$ | 17.275 | 0.919 | 1711.196 | 3.752 |
| $\lambda_1 = \lambda_2 = 0.5, \lambda_3 = \lambda_4 = 0.01$ | *17.560* | 0.923 | *1611.588* | 3.461 |
| $\lambda_1 = \lambda_2 = 0.5, \lambda_3 = \lambda_4 = 0.1$ | 16.623 | 0.898 | 1919.451 | 3.930 |
| $\lambda_1 = \lambda_2 = 0.5, \lambda_3 = \lambda_4 = 0.5$ | 17.394 | 0.920 | 1720.743 | 3.406 |

*4.4. Experimental Discussion*

From the above comparison of the subjective evaluation and objective metrics, we can see that the method presented in this paper is a significant improvement compared to existing methods. This paper's proposed method could be applied to various practical applications. For example, the method can effectively improve the brightness of medical images (ultrasound images, X-rays, etc.) and provide more accurate information. It can be applied to biological experiments to obtain images with high signal-to-noise ratios from low-expression samples, and it can also be applied to surveys of mines, mining caves, etc.

## 5. Limitations and Discussion

The method proposed in this paper is not perfect and has some shortcomings. As can be seen in Figures 7–9, despite being regarded as a better alternative, the suggested method still produces distorted images. This issue was demonstrated in Figure 10. Figure 10 shows a comparison with different parameters, from which it can be seen that the blue part of the folder area in the image differs from the real image, although the rest of the image is the same color, this is still something that needs to be investigated in the future. Figure 10 shows a comparison with different parameters, from which it can be seen that the blue part of the folder area in the image differs from the real image, although the rest of the image is the same color. This is still something that needs to be investigated in the future. As deep learning evolves, creating better networks to solve existing enhancement problems is a future focus of our research.

## 6. Conclusions

The paper presents a framework that aims to enhance low-light images using the Retinex theory. The proposed framework is composed of two components: the Decom-Net and the Enhance-Net. The first one receives the low-light image and obtains the necessary reflection and illumination images. These images are then fed into the Enhance-Net to enhance the illumination images. The decomposed reflection images are fused with the corrected illumination images to obtain the final output. The proposed network model incorporates residual connectivity and attention mechanisms to reduce image detail loss. It also includes a positional pixel attention mechanism to ensure that the enhanced network does not lose image detail features in the image illumination. Additionally, a constant color loss is proposed to correct possible color bias in the enhanced image. The results of the experiments revealed that the suggested method performed well in capturing subjective visual effects and evaluating different metrics. Compared with RetinexNet, the proposed method increased the PSNR, SSIM, and LPIPS by 4.6%, 1.8%, and 10.8% in the full-reference evaluation index, respectively, and increased the NIQE without reference by an average of 17.3%. Although the proposed method has been shown to improve the color perception of images, further research is needed to develop it.

**Author Contributions:** Conceptualization, J.W.; methodology, J.W. and H.W.; software, J.W. and H.W.; validation, Y.S. and J.Y.; investigation, J.W. and Y.S.; writing—original draft preparation, J.W. and H.W.; writing—review and editing, Y.S. and J.Y. All authors have read and agreed to the published version of the manuscript.

**Funding:** This research received no external funding.

**Institutional Review Board Statement:** Not applicable.

**Informed Consent Statement:** Not applicable.

**Data Availability Statement:** The data presented in this study are openly available online: https://blog.csdn.net/u014546828/article/details/109256136 (accessed on 24 November 2020).

**Conflicts of Interest:** The authors declare no conflict of interest.

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
