# Peer review of "Improved Retinex-Theory-Based Low-Light Image Enhancement Algorithm"

_applsci, doi:10.3390/app13148148_

Round 1
Reviewer 1 Report (New Reviewer)
In the paper, the authors propose a new algorithm for image enhancement in low light conditions. The algorithm is based on the use of two well-known neural networks and an additional loss function. The authors give a fairly comprehensive literature review and a fairly detailed description of their algorithm. At the end of the paper, they provide a detailed comparison of the processing results of their algorithm and its predecessors on three datasets. I think the paper is well written, the authors have achieved a concrete result with novelty and have clearly demonstrated its advantage over other algorithms. I can't say that the proposed algorithm has a total advantage, but the improvement in image processing performance over its predecessors can be clearly seen from the results presented. I think this article will be interesting and useful to the readers.
There are only two comments on the article:
1. It is probably worth giving brief characteristics of the datasets used: number and size of images, conditions under which they were acquired, etc.
2. It is worth giving information about the performance of the developed algorithm, preferably with a comparison with other algorithms used. This information will be useful for the readers to choose one or another algorithm for their tasks.
Author Response
Please see the attachment.

Reviewer 2 Report (New Reviewer)
As the authors describe in the introduction section, low-light optical imaging has a wide range of application needs in intelligent driving, military reconnaissance, etc. This paper proposes a novel network framework for low-light image enhancement based on retinex theory.
I think the main innovation of this paper is its effectiveness in reducing the loss of image details. Therefore, the method should also be applicable to biomedical applications, such as low-light fluorescence image enhancement. The paper is well written and organized. I have a few minor concerns.
1. I suggest that the authors discuss the effect of computer graphics card performance on the training speed of neural networks.
2. I suggest that the authors add the potential application outlook of the proposed method in the field of biomedical imaging in the discussion section of the manuscript. Fluorescence imaging with low light excitation has important potential applications in reducing phototoxicity and extending the acquisition time of biological samples. It is suggested to cite the following references on GPU acceleration to increase the speed of image processing. (Advanced. Photonics 4, 026003, 2022. Nat. Commun. 10, 1–11,2019 & The Innovation 4(3): 100425, 2023)
no comment
Author Response
Please see the attachment.

Reviewer 3 Report (New Reviewer)
1. Add the overall accuracy of the proposed method in the Abstract and Conclusion Sections.
2. Pseudocode / Flowchart and algorithm steps need to be inserted.
3. Time spent need to be measured in the experimental results.
4. Discussion and Limitation Sections need to be inserted.
Round 2
Reviewer 3 Report (New Reviewer)
Accept.
This manuscript is a resubmission of an earlier submission. The following is a list of the peer review reports and author responses from that submission.
Round 1
Reviewer 1 Report
This paper develops a low-illumination image enhancement framework. Simulation results demonstrate the superiority of the proposed approach against the conventional deep learning-based methods. However, it is believed that the contribution of this paper is limited. Please pay attention to the following comments.
1). It is better to state the main challenging problems to be solved in this paper.
2). The motivation for using pixel attention mechanism is unclear. Please specify it.
3). In page 6, the authors give the loss functions which involve lots of parameters. How to carefully choose the initial values of these parameters during the simulations? Please specify it.
4). It is believed that the given simulation results as well as its performance analysis are not enough to support the effectiveness of the proposed approach.
Reviewer 2 Report
The paper presents a low-light image enhancement model for which I have the following comments:
1. The English needs to be revised, multiple phrases are inconsistent. (ex. line 51-54 and line 31)
2. The materials and methods section should include details of the materials and methods of the proposed work. Yet it includes a literature review of very old literature.
3. The work lacks a proper review of the recent literature on the low light image enhancement techniques since the references are quite out-dated. You need to have a review for the work done during 2022 and 2023 mostly.
4. Line 137 mentions the Decom-Net model you propose. How is this different from the article presented in "https://link.springer.com/article/10.1007/s00521-019-04501-5".
5. How about the enhance-net. Is this used from a previous literature or you modified it? If it is similar to previous work, why is this work not referenced in section 3.2.
6. The LOL dataset presented in line 200 is not referenced as it should be as well.
7. The idea of choosing a specific image for experimentation is biased. How can you prove that the performance of your proposed model is the same on all other images from the datasets?.
8. Again in the experimentation section, the choice of 10 images to compare the different models to is not convenient. You need to have an average 10 fold cross validation to assure that the model really outperforms the other models in the different methods. Unless the dataset has a predefined train/test split which I don't think that this is the case since you altered the original dataset with synthetic dataset
9. you mention using the LOL dataset for training, then selecting 10 images from it for testing. Where this images used in the training as well?
An overall comment, the work seems to be outdated unless supported by more recent work (since as a quick count from the 30 references you used. None are from 2023, the most recent reference is only one from 2022, 7 from 2021 and quite a few from before 2010
Round 2
Reviewer 1 Report
The reviewer raises the following comments:
1) The reasons for initializing the parameters of the loss functions in formulas (1) and (2) are still hard-to-convince, although the authors claim that the selected experience value can achieve good results in the reply letter.
2) In the simulation part, the authors provide supplement experiments by merely comparing the quantitative of the LOL dataset across the PSNR, SSIM, MSE and LPIPS. It is still too simple and not enough in my view. In addition, how about the training loss of the improved loss function in formula (5)? Please consider the other performance metrics as well.
Reviewer 2 Report
The article has improved yet the English language still needs revisions.
Concerning the enhance net, you mentioned that "The enhance-Net is proposed by reading a lot of literature, without specific reference to a method or a specific literature, so there is no specific reference to the literature." This is quite ambiguous as it has to have an origin. Check this out "https://arxiv.org/abs/1612.07919"The results show minimum improvement versus the other methods thus the real benefit from the proposed method is not identified
Round 3
Reviewer 2 Report
article improved